# Genetically Modified Legume Plants as a Basis for Studying the Signal Regulation of Symbiosis with Nodule Bacteria

Andrey D. Bovin, Alexandra V. Dolgikh [ID], Alina M. Dymo, Elizaveta S. Kantsurova [ID], Olga A. Pavlova and Elena A. Dolgikh *[ID]

Laboratory of Signal Regulation, All-Russia Research Institute for Agricultural Microbiology, Podbelsky Chausse 3, 196608 St. Petersburg, Russia; dymoalina@yandex.ru (A.M.D.); oa.pavlova@arriam.ru (O.A.P.)
* Correspondence: dol2helen@yahoo.com; Tel.: +7-812-476-24-96

**Abstract:** The development of legume–rhizobial symbiosis results in the formation of nitrogen-fixing root nodules. In response to rhizobial molecules, Nod factors, signal transduction is mediated by the interaction of activated receptors with downstream signaling proteins. Previously, some new regulators of the signal pathway, such as phospholipases D, which regulate the level of phosphatidic acid (PA), as well as mitogen-activated protein kinases (MAPKs), have been identified in legumes. Since PA is an important signal messenger, we tested the hypothesis that increasing the level of proteins involved in the reversible binding of PA in plant tissues may have a positive effect on symbiosis. Our findings showed that overexpression of *MtSPHK1-PA*, encoding the PA-binding domain of sphingosine kinase 1 (SPHK1), stimulated plant growth and nodule development in legume plants. Furthermore, the influence of MAPK6 on the development of symbiosis was studied. Using genetic engineering methods, we increased MAPK6 transcriptional activity in transgenic roots, leading to an increase in the number of nodules and the biomass of pea plants. Therefore, new approaches to obtain plants with an increased efficiency of symbiosis were tested. We report here that both genes that encode signaling proteins may be used as potential targets for future modification using biotechnological approaches.

**Keywords:** signal transduction; phosphatidic acid; MAP kinase; gene expression; transgenic plants; *Medicago truncatula*; pea *Pisum sativum* L.





## 1. Introduction

One of the important tasks in the study of legume–rhizobia symbiosis is to increase its efficiency, which affects plant productivity. The development of new biotechnological approaches aimed at creating plants with increased functional activity of key regulators of nitrogen-fixing symbiosis development (mainly regulators of signal exchange between partners, which can affect the efficiency of this process) will solve this problem.

The formation of legume–rhizobia symbiosis is based on a signal exchange between partners, which leads to the formation of root nodules in which molecular nitrogen is fixed. Under the influence of rhizobial molecules, Nod factors, signal transduction from membrane lysin motif receptor-like kinases (LysM-RLK) [1–6] is mediated by the interaction of activated receptors with downstream signaling proteins [7,8]. Studying the signaling pathways involved in the regulation of symbiosis enables the possibility of searching for target genes that affect the efficiency of this process (the number of nodules, nitrogen-fixing activity, and morphometric parameters of plants). At the same time, signaling regulators may be potential targets for controlling the efficiency of legume–rhizobia symbiosis, which may be associated with increasing their functional activity, including genome editing methods.

Signal transduction triggers processes that lead to the development of infection threads with the help of which rhizobia colonize root tissues, as well as to the activation of nodule

organogenesis in root tissues remote from the surface [9]. A number of regulators of the Nod-factor-triggered signal transduction pathway have been identified in the model legumes *Lotus japonicus* and *Medicago truncatula*, such as receptor kinases with leucine-rich repeats (LRR-RK) LjSYMRK/MtDMI2 [10,11], calcium channels (LjPOLLUX/MtDMI1) [12–14], nuclear pore components (LjNUP85/133, LjNENA) [15–17], and calcium ATPase (LjMCA8) [18], which are necessary for the generation of periodic calcium oscillations in the nucleus and activation of calcium-dependent kinases (LjCCaMK/MtDMI3) [19,20] and transcription factors (CYCLOPS/IPD3, NSP1, NSP2, NIN, ERN1, NF-Y) [21–26]. However, most of the known signaling proteins are located either in the plasma membrane or in the nucleus, while little is known about how the signal is distributed throughout the cytoplasm from activated LysM-RLKs. This requires searching for the unknown proteins involved in signal transduction.

Previously, using proteomics and transcriptomic methods, we were able to show that in the early stages of signal transduction activated by Nod factors, a number of potential regulators of signal pathways can be activated. Among them are NADPH oxidase, lipoxygenase, heterotrimeric G-protein, phospholipases C and D, as well as mitogen-activated protein kinases (MAP kinases). However, since calcium plays a key role as a second messenger in signal transduction, it was of particular importance to further study intracellular regulators such as heterotrimeric G-proteins and phospholipases C and D involved in the formation of phospholipid derivatives (phosphatides). Phosphatides such as phosphatidic acid (PA), inositol-3 phosphate (IP3), and diacylglycerol (DAG) can influence the release of calcium from intracellular stores, as was shown in animals [27,28]. Another important system is the MAP kinase cascade, which typically consists of three protein kinases: a MAP kinase kinase kinase (MAPKKK), a MAP kinase kinase (MAPKK), and a MAP kinase (MAPK), which are involved in sequential activation by phosphorylation [29,30]. We hypothesized that increasing the rate of these particular processes may affect the efficiency of symbiosis.

Unlike in animals, which have a special class of G-protein-coupled receptors, in plants, heterotrimeric G proteins exist as separate complexes consisting of alpha, beta, and gamma subunits [31–33]. When the G protein is activated, the G protein alpha subunit is separated, and the G protein beta–gamma subunit complex is directly involved in signal transduction into the cell. In legumes such as *M. truncatula* and pea *P. sativum*, one beta subunit of the G protein (Gbeta1) was identified, and our experiments to reduce the expression of this gene using RNA interference, as well as experiments to localize the promoter activity of the *Gbeta1* gene, showed its necessity for signal transduction and control of symbiosis development [34]. Furthermore, we were able to identify several phospholipases C (PLC) and D (PLD) that interacted with the beta and alpha subunits of the G protein [34] and, apparently, similar to what is observed in some other processes in plants [35,36], may be involved in the formation of phosphatides such as PA and DAG. In fact, the content of these phosphatides was shown to increase in response to inoculation in legume plants [28,37–39]. Among them, phosphatidic acid (PA) is an important signal messenger and plays an essential role in fungi, animals, and plants during cell signaling, cell growth, development, and stress responses. PA is made up of glycerol-3-phosphate and two fatty acid chains [37,40], which are generated by activating PLD in the plasma membrane in response to specific signals. Rhizobial signal molecules, Nod factors, may stimulate the activation of PLD and PLC during the development of symbiosis.

Furthermore, we hypothesized that if the accumulation of PA in legume plants increases, this may have a positive effect on symbiosis. PA-binding proteins have been well studied in plants such as Arabidopsis [41]. Knowledge of the active site that binds to PA may be helpful in identifying proteins that contain these sites in legume plants.

Other important signal regulators are MAP kinases such as mitogen-activated protein kinases (MAPKKs) and their substrate mitogen-activated protein kinases (MAPKs) in legumes [42,43]. Experiments that aimed to decrease MtMAPKK4/LjSIP2 and MtMAPK6/LjMAPK6 using RNA interference in the model legumes *M. truncatula* and *L. japoni-*

*cus* resulted in a significant decrease in infection thread number and symbiotic nodule amount [42–44]. In *L. japonicus*, MAPK6 has recently been found to compete with histidine phosphotransfer protein 1 (LHP1) to bind to the receiver domain of the cytokinin receptor (histidine kinase 1, LHK1) and suppress the expression of two genes of 1-aminocyclopropane-1-carboxylic acid synthases involved in the production of ethylene, which is a negative regulator of nodule formation [44]. Therefore, MAPK6 may have a positive effect on the development of symbiosis. In fact, the overexpression of the homologous *MsSIMK* gene in the crop legume alfalfa *Medicago sativa* was related to stimulation of nodulation and plant growth parameters such as shoot length and biomass [45].

In our studies, we focus on two types of potential regulators of the signal transduction pathway, such as PA-binding proteins (SPHK1) and MAP kinases (MAPK6). To analyze the effect of these regulators on the development of symbiosis and the efficiency of nodule formation, in this experimental work, composite plants of *M. truncatula* and *P. sativum* have been created with overexpression of the *SPHK1* gene fragment that encodes the PA binding domain and MAPK6 in transgenic roots and nodules. The number of nodules and morphometric parameters of plants have been estimated to show their influence on the efficiency of symbiosis.

## 2. Materials and Methods

### 2.1. Bacterial Strains and Conditions for Their Cultivation

The *Escherichia coli* strain TOP10 (Thermo Fisher Scientific, Waltham, MA, USA) was used for gene cloning. The selection of transformants was carried out in LB agar medium with the addition of appropriate antibiotics (kanamycin 100 µg/mL for pDONR221, spectinomycin 50 µg/mL for pB7WG2D) at 37 °C. *Sinorhizobium meliloti* 2011 and *Rhizobium leguminosarum* bv. *viciae* RCAM 1026 were used for plant inoculation. They were grown in liquid B$^-$ medium [46] at 28 °C in a flat bottom flask on a rotary shaker. The *Agrobacterium rhizogenes* Arqua 1 strain was grown in TY agar medium with the addition of spectinomycin 50 µg/mL at 28 °C.

### 2.2. Plant Growth Conditions

The *Medicago truncatula* cv Jemalong A17 plants were grown in vermiculite moistened with Fahraeus medium at +21 °C, 60% humidity, and 16 h day/8 h night period. Plants were inoculated with a suspension of the *S. meliloti* 2011 strain with a density of OD$_{600}$ = 1.0 with 2 mL per plant. Plant material was collected 14 days after inoculation. The pea plants were grown in vermiculite watered with Jensen medium at +21 °C, 60% humidity, and 16 h day/8 h night period. Plants were inoculated with a suspension of the *R. leguminosarum* bv. *viciae* RCAM1026 strain with a density of OD$_{600}$ = 1.0 with 2 mL per plant. Plant material was collected 21 days after inoculation.

### 2.3. Molecular Cloning

2.3.1. Cloning of the MtSPHK1 Gene Fragment Encoding Domain for Phosphatidic Acid Binding

The cDNA of *Arabidopsis thaliana* sphingosine kinase1 (At4g21540) [41] was used for searching the homologous genes in *M. truncatula* and *P. sativum* genomes [47,48]. It resulted in the identification of the *MtSPHK1* (MtrunA17_Chr2g0281181) and *PsSPHK1* (Psat1g208320) homologs. A fragment (723 b.p.) of the coding sequence of the *MtSPHK1* gene was amplified using a high-fidelity polymerase Phusion 548S (Thermo Fisher Scientific, Waltham, MA, USA) and specific primers (Table S1). The amplified fragment was subcloned into the pDONR221 vector using BP clonase (Thermo Fisher Scientific, Waltham, MA, USA), and finally into the pB7WG2D destination vector under a constitutive promoter pCaMV35S (p35S) using LR clonase (Thermo Fisher Scientific, Waltham, MA, USA).

### 2.3.2. Cloning of the MAPK6 Gene

The coding sequence of the *PsMAPK6* gene (Psat7g119280) was amplified with primers containing the *attB1* and *attB2* sequences using Phusion 548S high-fidelity polymerase (Thermo Fisher Scientific, Waltham, MA, USA). The amplified fragment was purified from agarose gel and cloned in the intermediate pDONR221 vector using BP clonase (Thermo Fisher Scientific, Waltham, MA, USA). The fragment was then transferred to the pB7WG2D destination vector using LR clonase (Thermo Fisher Scientific, Waltham, MA, USA). The final constructs were transferred to *A. rhizogenes* Arqua 1 for plant transformation.

### 2.4. Transformation of Pea Pisum sativum Plants

Pea *P. sativum* L. seeds cv. Frisson were sterilized with sulfuric acid for 5 min, followed by washing at least 4 times in excess of sterile water. Seeds were grown on 1% water agar for 4–5 days at 23 °C in the dark. The young seedlings were transferred to light in liquid Jensen medium [49], and grown for 5–7 days (+21 °C, 60% humidity, 16 h day/8 h night) until one internode formed. Seedlings were transformed by applying a suspension of *A. rhizogenes* Arqua 1 bacteria carrying the appropriate construct to the cut surface in the hypocotyl region. After transformation, the plants were placed in a Jensen solid medium and incubated for 10–11 days until callus appeared [50]. After this, the plants were transferred to emergence medium [51] containing cefotaxime (150 µg/mL) and incubated for another 3–5 days until transgenic roots appeared. The transgenic roots were selected on the basis of the presence of the fluorescent reporter proteins DsRED or GFP in the cells. Plants with transgenic roots were transferred into the pots with vermiculite moistened with Jensen medium and grown in a growth chamber at +21 °C, 60% humidity, and a 16 h day/8 h night cycle.

### 2.5. Transformation of Medicago truncatula Plants

The seeds of the Jemalong A17 *M. truncatula* line were sterilized in concentrated sulfuric acid for 10 min, washed with sterile water, then incubated in bleach for 2 min, and washed with sterile water. The seeds were transferred to 0.8% water agar in a Petri dish and kept at 4 °C for 24 h, after which the dish was transferred to the dark at 23 °C for seed germination overnight. The seedlings were transformed 48 h after germination by applying a suspension of *A. rhizogenes* Arqua 1 bacteria carrying the required construct to the cut surface in the hypocotyl region. The plants were incubated in Fahraeus medium for 7 days until callus appeared and transferred to emergence medium with cefotaxime (300 µg/mL) for 7–10 days until transgenic roots appeared. Plants with transgenic roots were placed into vermiculite moistened with Fahraeus medium.

### 2.6. Isolation of Total RNA

Total RNA was isolated from *M. truncatula* and *P. sativum* roots and nodules, which were ground with a mortar to a powder in liquid nitrogen and extracted with Trizol reagent (Bio-Rad Laboratories, Hercules, CA, USA). DNA was eliminated using DNAse digestion (Thermo Fisher Scientific, USA). RNA quality and quantity were determined by Nanodrop (Implen GmbH, Munich, Germany). 1 µg of total RNA was used for cDNA synthesis using 200 U of RevertAid H Minus reverse transcriptase (RT) (EP0451, Thermo Fisher Scientific, USA) in 20 µL of 50 mM Tris-HCl (pH 8.3 at 25 °C), 50 mM KCl, 4 mM $MgCl_2$, 10 mM DTT reaction mixture with 1 µL of 25 mM dNTPs. To exclude DNA contamination in the template RNA, the negative controls for RNA samples in the mixture without RT (RT-control) were also prepared. The cDNA samples were diluted to a total volume of 100 µL.

### 2.7. Quantitative Reverse Transcription-Polymerase Chain Reaction (RT-qPCR) Analysis

To analyze the *MtSPHK1* and *PsMAPK6* gene expression level in transgenic plants, quantitative reverse transcription-polymerase chain reaction (RT-qPCR) was carried out using SYBR Green qPCR Supermix (Evrogen, Moscow, Russia) according to the man-

ufacturer's recommendations and performed on a CFX Opus 96 PCR system (Bio-Rad Laboratories, Guangzhou, China) with initial denaturation at 95 °C for 3 min followed by 40 cycles of 95 °C for 30 s, 54–58 °C for 40 s and 72 °C for 30 s. Melting curve analysis was performed from 54 to 95 °C, where the temperature increased by 0.5 °C every 5 s. The total volume was 10 µL per sample, containing 1 µL of cDNA and 2 µL of 5xSYBR Green qPCR Supermix (Evrogen, Moscow, Russia). Highly specific primers were designed using Vector NTI with melting temperatures (Tm) between 54 and 58 °C and amplicon lengths of 100–300 bp. As references, the housekeeping gene of the *Ubiquitin* for pea and *40S* for *M. truncatula* was used (Table S1). The threshold cycle (Ct) values were calculated using Bio-Rad CFX Maestro 2.3 software (Bio-Rad Laboratories, Guangzhou, China) and analyzed using the $2^{-\Delta\Delta Ct}$ method [52].

### 2.8. Statistical Data Analysis

Statistical data processing was carried out using Microsoft Office Excel 2016 (Microsoft Inc., Redmon, WA, USA). The significance of the differences between the values was assessed using analysis of variance (ANOVA).

## 3. Results

### 3.1. The Influence of MtSPHK1 Gene Fragment Encoding Domain for Phosphatidic Acid Binding in Symbiosis

Since phosphatidic acid (PA) is an important signal messenger, it was hypothesized that its accumulation in legume plants may have a positive effect on symbiosis. PA may act as a signaling molecule and regulate the activity of target proteins, the structure of the membrane, and vesicular trafficking [53]. To estimate the influence of this lipid mediator, it was suggested to increase the level of proteins involved in the reversible binding of PA in plant tissues. Recently, searching for plant proteins that can bind to PA resulted in their identification in Arabidopsis and rice [41,54]. Among them, the sphingosine kinase 1 (SPHK1) contains a classical PA binding site, VSGDGI [41].

In the first stage, the homologous *MtSPHK1* and *PsSPHK1* genes were found in *M. truncatula* and *P. sativum* genomes [47,48]. Experiments were initiated in the model legume plant *M. truncatula*. A fragment of the *M. truncatula* sphingosine kinase 1 gene (723 b.p.), encoding the PA-binding domain of the protein (MtSPHK1-PA) without the catalytic domain (Figure S1), was used for subsequent experimental work. To study the functional role of this PA-binding domain in the regulation of symbiosis, *M. truncatula* composite plants with overexpression of *MtSPHK1-PA* under the p35S promoter (*MtSPHK1-PA-OE*) were created using transformation with *Agrobacterium rhizogenes*. The level of *MtSPHK1-PA* expression was shown to increase approximately 40 times in nodules in *MtSPHK1-PA-OE* plants compared to controls overexpressing the *beta-glucuronidase (GUS)* gene (*GUS-OE*) (Figure 1).

Overexpression of *MtSPHK1-PA* resulted in a 1.5–2 times increase in the number of nodules in *MtSPHK1-PA-OE* plants compared to *GUS-OE* control plants. At the same time, the effect of *MtSPHK1-PA* overexpression on plant growth was also estimated. A significant increase in the length of the shoot and roots was also found in *MtSPHK1-PA-OE* plants in our experiments (Figure 1). In addition to this, a small but significant increase in total biomass was found in plants with stimulation of PA binding (*MtSPHK1-PA-OE*) compared to *GUS-OE* control plants. Therefore, an increase in the accumulation of PA, which is a secondary messenger during signal transduction in response to Nod factors, in the root of legume plants can have a significant positive effect on the development of symbiosis. Since a stimulating effect of *MtSPHK1-PA* overexpression on the efficiency of symbiosis was found in the model legume *M. truncatula*, similar experiments were initiated in the crop legume *P. sativum*, which also forms an indeterminate type of nodules as *M. truncatula*. Future experiments will demonstrate whether the influence of increased PA levels has a similar effect in both legumes.

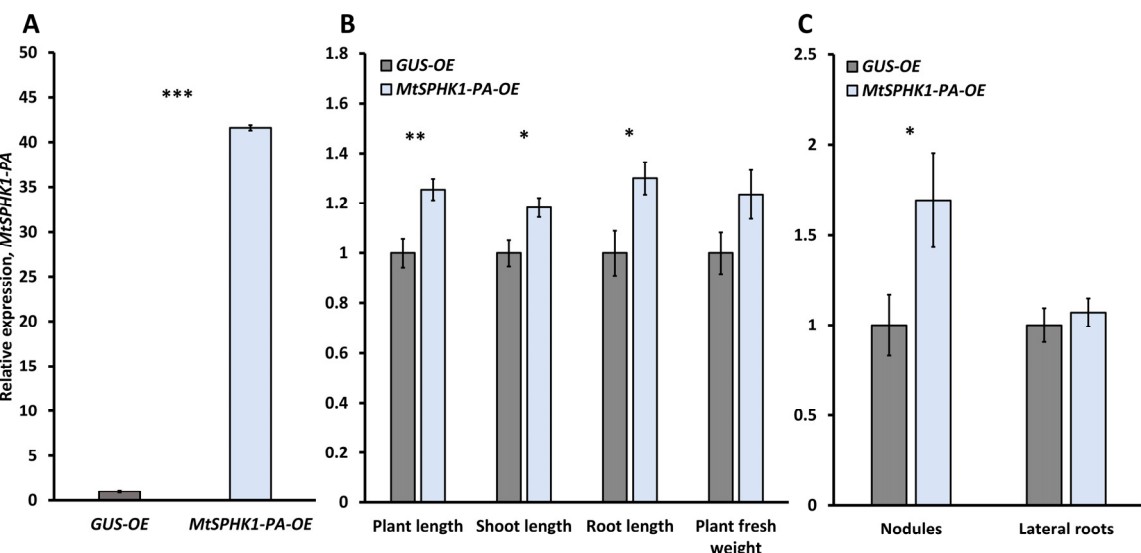

**Figure 1.** Comparative analysis of the *MtSPHK1* gene expression, growth parameters, number of nodules, and lateral roots in *MtSPHK1-PA-OE* and *GUS-OE* plants. The *MtSPHK1* gene expression (**A**). As a control, the nodules of plants overexpressing the *beta-glucuronidase (GUS)* gene (*GUS-OE*) were used. The expression was normalized against the constitutively expressed *40S* gene. The transcript level in nodules of *GUS-OE* plants was set to 1 (control), and the level in nodules of *MtSPHK1-PA-OE* plants was calculated relative to the control values. Analysis of changes in gene expression was carried out on the basis of two biological repeats. The data for 10–15 samples were analyzed ± standard error of the mean (SEM). The asterisks indicate statistically significant differences based on a one-way analysis of variance (one-way ANOVA). The plant shoot and root lengths, and plant fresh weight (**B**). Number of nodules and lateral roots (**C**). The values in *GUS-OE* plants were set to 1 (control), and the values in *MtSPHK1-PA-OE* plants were calculated relative to the control. *, Significant difference at $p \leq 0.05$; **, Significant difference at $p \leq 0.01$; ***, Significant difference at $p \leq 0.01$.

### 3.2. The Effect of MAPK6 Overexpression on the Development of Symbiosis

In addition, the effect of *MAPK6* overexpression on the development of symbiosis was also estimated. Experiments were carried out on pea *P. sativum*, an important agricultural crop. Composite plants with *PsMAPK6* overexpression were created and compared to *GUS-OE* controls. The level of *PsMAPK6* expression was shown to increase approximately 6–7 times in roots and nodules in *PsMAPK6-OE* plants compared to *GUS-OE* controls (Figure 2). Our findings showed an increase in the number of nodules and the total biomass of plants with upregulated *PsMAPK6* compared to the *GUS-OE* control (Figure 2). At the same time, the local clustering of nodules in groups consisting of 4–5 nodules (probably as a result of local suppression of ethylene production) was observed (Figure 3). Therefore, overexpression of *PsMAPK6* in peas promotes nodule number and clustering, as well as plant biomass production. Finally, our experiments showed that MAPK6 may be another important potential target for the regulation of symbiosis efficiency.

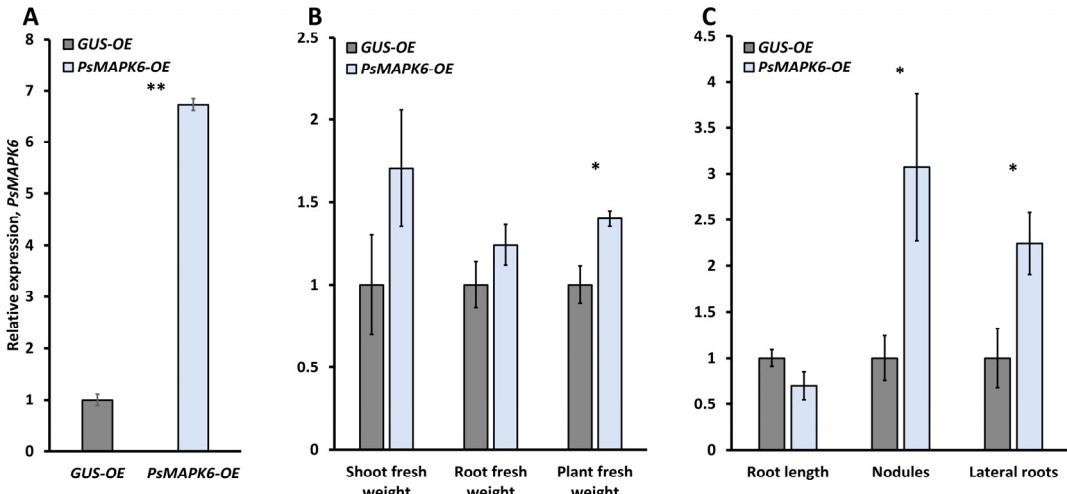

**Figure 2.** Comparative analysis of the *PsMAPK6* gene expression, growth parameters, and number of nodules and lateral roots in *PsMAPK6-OE* and *GUS-OE* plants. The *PsMAPK6* gene expression (**A**). As a control, the nodules of plants overexpressing the *beta-glucuronidase (GUS)* gene (*GUS-OE)* were used. The expression was normalized against the constitutively expressed *Ubiquitin* gene. The transcript level in nodules of *GUS-OE* plants was set to 1 (control), and the level in nodules of *PsMAPK6-OE* plants was calculated relative to the control values. The data for 5–6 samples were analyzed ± standard error of the mean (SEM). The asterisks indicate statistically significant differences based on a one-way analysis of variance (one-way ANOVA). The plant shoot and root lengths, and plant fresh weight (**B**). Number of nodules and lateral roots (**C**). The values in *GUS-OE* plants were set to 1 (control), and the values in *PsMAPK6-OE* plants were calculated relative to the control. The RT minus controls showed no DNA contamination in the template RNA (Figure S2). *, Significant difference at $p \leq 0.05$; **, Significant difference at $p \leq 0.01$.

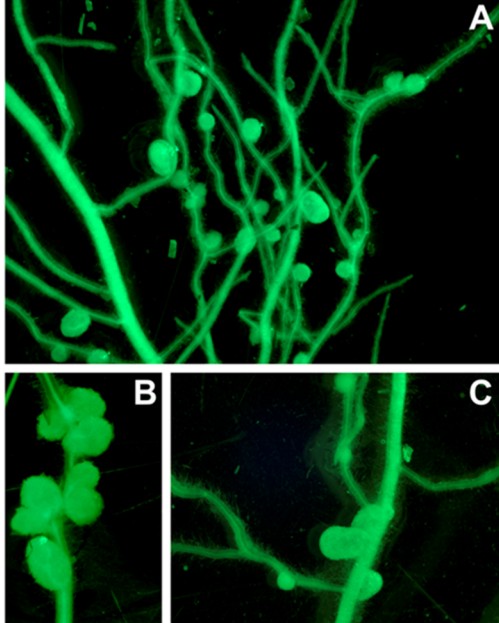

**Figure 3.** The effect of *PsMAPK6* overexpression on the development of nodules in *PsMAPK6-OE* plants (**A**,**B**) compared to *GUS-OE* controls (**C**).

## 4. Discussion

The cellular level of PA is dynamically regulated in response to various stimuli in plants [53,55]. The accumulation of this phosphatide is generated by activating PLD

in the plasma membrane in response to specific signals. In addition, another enzyme, diacylglycerol kinase (DAGK), can phosphorylate diacylglycerol (DAG) to form PA [56], but PLD is generally considered the key enzyme that regulates PA synthesis. PLD was shown to be activated and the PA content increased transiently in legume roots after rhizobial infection [28,34,37,38]. Our findings showed that overexpression of *MtSPHK1-PA* encoding the PA-binding domain of the SPHK1 enzyme stimulated plant growth and nodule development in legume plants. These results were in agreement with data from experiments on the inhibition of PA production by n-butanol (inhibition of PLD involved in PA synthesis) that reduced the number of nodules in legumes [28,38]. This suggests that the PA level is transiently increased in *MtSPHK1-PA*-OE plants. As a result of fatty acid chains in the PA molecule, the formation of micelles can occur in the cell in response to an increase in its content. Changes in biophysical properties can result in diverse functions.

The influence of various PLDs and PA has been extensively studied for their involvement in plant responses to various abiotic and biotic stresses, such as interactions with pathogenic or beneficial microorganisms, heat stress response, and hormone signaling [57,58]. In legumes, the role of PA has been investigated in actin and tubulin cytoskeleton dynamics during rhizobial infection and nodulation in soybean [37]. These studies showed that PA as a signaling mediator is capable of binding to a number of target proteins and is involved in membrane structure modulation. The membrane structure modification takes place during infection thread formation and rhizobial release from these tubular structures during symbiosis development. In addition, it may be suggested that PA derived by PLD can influence the release of calcium from intracellular stores during nodulation through interaction with membrane protein channels that stimulates increased cytosolic calcium level during signal transduction. This suggestion may be verified in our future experiments. Therefore, an increase in PA accumulation, which is a secondary messenger during signal transduction in response to Nod factors, in legume plants can have a significant positive effect on the development of symbiosis.

The effect of overexpression of MAP kinases on symbiosis development was also estimated. Our findings suggest that *MAPK6* promotes the number of nodules and their clustering, as well as the production of plant biomass. Similar results have been obtained in alfalfa *M. sativa* with overexpression of *MsSIMK*, which is homologous to *PsMAPK6* [45]. This means that in both legumes with similar indeterminate types of nodules, MAPK stimulation may be considered as a suitable target to increase the efficiency of symbiosis.

These data suggest that genetic manipulations that cause overexpression of genes that encode PA-binding proteins such as MtSPHK1-PA and MAPK6 positively affect the development of shoots and roots, as well as the formation of nodules in both legumes (*M. truncatula* and *P. sativum*), that points to the new biotechnological potential of these proteins. Therefore, both genes encoding these proteins may be used as potential targets for future modification using biotechnological approaches.

## 5. Conclusions

Searching for new biotechnological approaches for plant crop improvement has great significance. Stimulation of metabolic and signal pathways in plants may help to perform this task. It was recently demonstrated that PLDs and MAPKs are involved in the signal regulation of legume–rhizobia symbiosis. Since the metabolic product of PLDs, PA, may play an important physiological role as a second messenger, the increase in intracellular PA appears to be essential for the regulation of target proteins during the development of symbiosis. Here, the overproduction of two types of potential regulators of the signal transduction pathway, such as PA-binding protein SPHK1 and MAPK6, was shown to stimulate plant growth and nodule development in *M. truncatula* and *P. sativum.* Due to their significant positive effect on the efficiency of symbiosis, both regulators may be used as potential targets for future modification using biotechnological approaches.

**Supplementary Materials:** The following supporting information can be downloaded at: https://www.mdpi.com/article/10.3390/horticulturae10010009/s1, Figure S1: Domain structure of the SPHK1 protein. PA—phosphatidic acid, (PA)—binding domain, CD—catalytic domain; Figure S2: RT minus control analysis; Table S1: List of primers used in this study.

**Author Contributions:** A.D.B., investigation, plant transformation, and data analysis; A.V.D., plant transformation and investigation; A.M.D., methodology and cloning; E.S.K., methodology and cloning; O.A.P., qPCR and cloning; E.A.D., conceptualization, writing and editing, and supervision. All authors have read and agreed to the published version of the manuscript.

**Funding:** This work was supported financially by the Russian Science Foundation RSF 21-16-00106.

**Data Availability Statement:** Data are contained within the article.

**Acknowledgments:** The research was performed using equipment of the Core Centrum "Genomic Technologies, Proteomics and Cell Biology" in ARRIAM.

**Conflicts of Interest:** The authors declare no conflict of interest.

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
