# Peer review of "Genetically Modified Legume Plants as a Basis for Studying the Signal Regulation of Symbiosis with Nodule Bacteria"

_horticulturae, doi:10.3390/horticulturae10010009_

Round 1

Reviewer 1 Report

Comments and Suggestions for Authors

The study in the submitted article aimed to identify new biotechnological approaches for creating plants with increased functional activity of key regulators of nitrogen-fixing symbiosis development. The authors investigated two types of regulators: PA-binding proteins (SPHK1) and MAP kinases (iMAPK6), and this analysis was carried out on the legume species: Medicago truncatula and Pisum sativum. They find positive and significant effects on the development of plant shoots, roots, plant biomass and the formation of nodules in these to species. The obtained results are significant for future experiments on modification using biotechnological approaches.

The introduction explains very well why this experiment was necessary to be carried out. The hypotheses are well established. Everything is clearly described. The material and methods are described in great detail. Everything is clear and sufficient information is provided about the setting of the experiment. Authors write in the first person (plural), and it is necessary to correct that.

The results are clearly displayed. The results have been correctly analysed. Also, some sentences are necessary to rearrange and clarify. In the discussion, the authors link to each segment of their experiment and supplement their findings with arguments. The conclusions are well stated.

The cited references are relevant to the experiment. Out of 55 cited references, 16 were published within the last 5 years (30%), 22 (40%) within the last 10 years (average within 12 years). Only three of 55 are auto citations (5, 34, 48), and they are relevant to the subject.

Specific comments:

Lines 31-35: Please move this sentence to the end of the Introduction, or rephrase it

Line 66: “it was of particular interest to us to further study intracellular regulators” – not only for you – Suggestion – “it is of great importance to further study intercellular regulators”

Line 95: “Furthermore, we hypothesized that if the accumulation of PA in legume plants” – Please state this hypothesis in general – e.g. It can be hypothesized that...

Line130: Missing letter i in word “Agrobacterium

Line 181: Please remove spaces between numbers and degrees throughout the text, e.g. replace “4 ° C” with “4°C”

Line 183: So far you've been writing this separately -  “Arqua1”

Line 185: Please replace the word calli with callus

Lines 212, 216, 233, 279: Please replace “we” with e.g. ”it was”

Line 242: Please use only one term pea or Pisum sativum L.

Lines 277, 282, 332, 346: Please replace the word “our” with the word “this”

Lines 324-325: Please replace the words “we suggest” with the phrase “it may be suggested”

Lines 327-328: Please replace “This suggestion will be verified in our future experiments.” with e.g. “This suggestion may be verified in a future experiment.

Line 331: Please replace “We have” with “it was”

Lines 346 and 347: Please write the names of the genes in italics

Author Response

Answer to Reviewer 1

We thank the Reviewer for the consideration of our work and helpful comments. The manuscript has been modified in accordance with your suggestions. Version of the manuscript with tracking changes has also been submitted.

Line 31 – 34. We have rephrased this sentence in the manuscript.

Line 65. The phrase has been modified.

Line 94. If the reviewer finds it possible, we would like to leave this phrase, because it was important suggestion.

Line 128, 163, 181, 182, 184. The corrections have been included.

Line 212, 216, 233, 270, 279, 321, 325, 329 – The sentences have been modified.

The conclusion has been changed in accordance with recommendation.  

Reviewer 2 Report

Comments and Suggestions for Authors

Dear Authors

I appreciate your idea and efforts. Some changes are needed to improve the manuscript.

1-Please improve the M&M it is a bit confusing. For example;

"2.1. Bacterial strains and conditions for their cultivation

Cloning was performed using......." 

It is not clear why you needed to do this cloning and what exactly cloned here. Bacteria itslef was cloned? or some gene in bacteria?

or in line 217; "In the first stage, the experiment was designed to search for proteins that... " it is not clear that in which step of M&M this part was performed. Please provide clearly in M&M each step was done for which aim. I give here an example:

In the M&M at 2.7 qRT-PCR analysis you can write for example to analyse expression level of ???? gene in transgenic Medicago (?) we performed qRT-PCR and the rest and provide the primers sequences that you used for gene expression analysis. There are no supplementary files available to download. 

Or

2.6. Isolation of total RNA from plant roots

It is not clear root of what? How many replicates did you have. Were there also control plants?

2- Once you used the full name and gave the acronym then you only use the acronym. For example; phosphatidic acid (PA) look at line 212, this is same for others too. Or you wrote  "pea P. sativum" no need to write " P. sativum" all the time.

3- You used "at the same time " a lot. for example I wouldn't start line 272 with "at the same time"

4- Conclusions part is too short and the line 338 to 343 are kinda duplication of Conclusions part. This parts needs to be rewritten.

Best wishes

Author Response

Answer to Reviewer

We thank the Reviewer for the interest in our work and the helpful comments and suggestions, which improved the manuscript. We also included a version of the manuscript with tracking changes.

The M&M have been modified.

Line 123. The sentence about cloning has been changed in accordance with your comment.

Line 143 – 147. We included the information about searching of genes in the M&M.

Line 193. This phrase was modified. The information about number of replicates and control plants was presented in Figure legends.

Line 200. The information about genes has been included.

We are sorry for inconvenience and included the supplementary files as the attached file.

Line 216. We have used the full name and the acronym again in this case, just to remind about this compound in the Results.

Line 220 – 224. We have tried to describe in detail the information about searching of homologous genes.

The conclusion has been changed in accordance with recommendations.  
